# Rheumatologic Perspectives on Sarcoidosis: Predicting Sarcoidosis-Associated Arthritis Through Comprehensive Clinical and Laboratory Assessment

**DOI:** 10.3390/jcm13247563

**Published:** 2024-12-12

**Authors:** Ali Ekin, Salim Mısırcı, Oğuzhan Sertkaya, Belkıs Nihan Coşkun, Burcu Yağız, Ediz Dalkılıç, Yavuz Pehlivan

**Affiliations:** 1Division of Rheumatology, Department of Internal Medicine, Faculty of Medicine, Bursa Uludag University, 16059 Bursa, Türkiye; dr.salim-misirci@hotmail.com (S.M.); belkisnihanseniz@hotmail.com (B.N.C.); burcuyilmaz_84@hotmail.com (B.Y.); edizinci@hotmail.com (E.D.); drypehlivan@gmail.com (Y.P.); 2Department of Internal Medicine, Faculty of Medicine, Bursa Uludag University, 16059 Bursa, Türkiye; oguzhansertkaya@uludag.edu.tr

**Keywords:** arthritis, 25-hydroxyvitamin D, diabetes mellitus, rheumatology, sarcoidosis, laboratory

## Abstract

**Background/Objectives:** The primary aim of our study was to evaluate the demographic, clinical, and laboratory characteristics of sarcoidosis patients with musculoskeletal symptoms; investigate the relationship between arthritis development and various laboratory parameters (such as vitamin D, liver enzymes, and ACE levels); and compare the sarcoidosis-associated arthritis cases with those without. We also explored the factors influencing arthritis development and the role of biopsy in diagnosing sarcoidosis within rheumatology practice. **Methods:** This retrospective study analyzed 147 sarcoidosis patients from 2000 to 2024, categorized by the presence (*n* = 45) or absence (*n* = 102) of arthritis. Demographic, clinical, and laboratory data, including biopsy results, were collected and compared. **Results:** The mean age was 56.02 ± 11.21 years, with a mean disease duration of 134.33 ± 56.98 months. Females constituted 86.4% of the cohort. All of the patients presented musculoskeletal involvement. Pulmonary involvement was present in 93.7% of cases, and extrapulmonary involvement included the skin (21.20%), the eyes (14.30%), and peripheral lymphadenopathy (10.6%). Methotrexate was the most common treatment after corticosteroids. In the arthritis group, diabetes mellitus was more frequent (*p* = 0.024), the GGT levels were higher (*p* = 0.044), and the 25-hydroxyvitamin D levels (*p* = 0.002) and the DLCO Adj (*p* = 0.039) were lower. Multivariable regression showed diabetes mellitus (*p* = 0.028, OR: 4.805, 95% CI: 1.025–22.518) and low 25-hydroxyvitamin D levels (*p* = 0.034, OR: 0.914, 95% CI: 0.841–0.993) as factors influencing arthritis development. The other parameters showed no significant differences. **Conclusions:** This study identified significant clinical, demographic, and laboratory differences between sarcoidosis patients with and without arthritis. The patients with sarcoidosis-associated arthritis exhibited a higher frequency of diabetes mellitus, lower levels of 25-hydroxyvitamin D, and elevated GGT levels. Additionally, the lower DLCO values in the patients with arthritis indicate a more severe impact on pulmonary function, underscoring the importance of comprehensive pulmonary evaluation in this subgroup.

## 1. Introduction

Sarcoidosis is a rare systemic disease of unknown etiology, characterized histopathologically by non-caseating granulomas [1,2]. It can involve multiple organs and systems, including pulmonary infiltrates, hilar lymphadenopathy, the skin, the eyes, and cardiac involvement [3,4]. The reported incidence and prevalence of sarcoidosis vary across ethnic groups and geographic regions. The annual incidence is estimated to range from 0.1 to 81 per 100,000, and the prevalence is estimated to range from 0.1 to 640 per 100,000 [5]. In Turkey, the incidence of sarcoidosis is reported to be 4 per 100,000 [5,6]. Sarcoidosis is more frequently observed in women, and the typical age of onset is between 20 and 40 years, with a second peak occurring in those over 50 years of age [1].

In up to 90% of sarcoidosis cases, lung involvement is observed, leading to significant advancements in the understanding and treatment of sarcoidosis that are largely attributable to the efforts of the pulmonology community. The classic Scadding staging system, developed half a century ago, evaluates pulmonary and intrathoracic involvement and is still widely used today [7]. However, it almost entirely overlooks extrapulmonary disease. With the increasing recognition of sarcoidosis as a systemic disease and the development of new therapeutic approaches, the importance of multidisciplinary treatment is growing [8]. In patients presenting to a rheumatology clinic, extrapulmonary manifestations, such as arthritis, vasculitic involvement, erythema nodosum, other skin lesions, eye involvement, and neurological and cardiac manifestations, may be observed [9]. The clinical presentation of sarcoidosis varies depending on the affected organ and system and can range from asymptomatic to fatal [2]. The prognosis depends on the onset, the pattern of involvement, and the severity. Sarcoidosis, with its multisystemic nature, can be confused with malignancy, infection, and various rheumatic diseases [10,11,12]. A diagnosis requires clinical and radiological findings, the demonstration of non-caseating granulomas in tissue samples, and the exclusion of other granulomatous diseases [13]. Elevated serum angiotensin-converting enzyme (ACE) and calcium levels, as well as an increased CD4/CD8 ratio in bronchoalveolar lavage (BAL), can aid in this diagnosis [14,15]. Sarcoidosis-associated arthritis (SAA) is influenced by various factors, including genetic predisposition, immune dysregulation, and systemic involvement. Genetic studies have linked certain HLA types, such as HLA-DRB1, to an increased risk of developing arthritis in sarcoidosis patients [2,3]. Systemic manifestations, particularly pulmonary and skin involvement, are also strong predictors of arthritis development. Patients with a higher burden of systemic disease, including pulmonary sarcoidosis, tend to show a higher incidence of joint involvement [3]. Sarcoidosis can affect vitamin D metabolism, leading to disruptions in calcium homeostasis and potentially resulting in hypercalcemia [3]. Vitamin D deficiency, commonly observed in sarcoidosis patients, may impair immune regulation and contribute to the development or exacerbation of inflammatory conditions, including musculoskeletal symptoms such as arthritis [2]. Previous studies have suggested that these metabolic alterations can intensify the severity of the musculoskeletal manifestations in sarcoidosis patients [2,3].

Sarcoidosis can present with numerous rheumatologic symptoms and findings. Musculoskeletal involvement is one of the most common reasons for patients to consult rheumatologists, and it is observed in approximately 15–25% of cases [16]. These clinical features can develop before or concurrently with pulmonary involvement. Consequently, patients may be monitored under diagnoses other than sarcoidosis, potentially delaying the diagnosis of sarcoidosis [17]. Sarcoidosis can mimic and coexist with diseases such as rheumatoid arthritis (RA), ankylosing spondylitis (AS), Sjögren’s syndrome, and vasculitis [10,18]. Therefore, rheumatologists must consider sarcoidosis in their differential diagnosis.

Our study had several important aims. Firstly, we aimed to evaluate the demographic, clinical, and laboratory characteristics of the patients with a diagnosis of sarcoidosis who developed musculoskeletal symptoms and the patients with musculoskeletal symptoms who were diagnosed with sarcoidosis as the result of a diagnostic investigation. In particular, this study aimed to investigate the relationship between the development of arthritis and various laboratory parameters, including vitamin D levels, liver enzyme levels, and ACE levels, as well as clinical features and comorbid conditions. A comparative analysis was conducted between the patients with sarcoidosis-associated arthritis and those without, focusing on the clinical and laboratory similarities and distinctions. Furthermore, this study aimed to underscore that not all cases of arthritis observed in sarcoidosis patients are attributable to the underlying disease and to investigate the factors contributing to the development of sarcoidosis-related arthritis from a rheumatologic standpoint. Additionally, we present our experience regarding the role of biopsy in diagnosing sarcoidosis within rheumatologic practice.

## 2. Materials and Methods

### 2.1. Study Design and Patients

In this study, we retrospectively reviewed the medical records of the patients diagnosed with sarcoidosis and followed up at our rheumatology clinic. The inclusion criteria were as follows: (1) age 18 or older; (2) classic clinical presentation (Löfgren’s or Heerfordt syndromes) with a high likelihood of sarcoidosis according to the 2022 WASOG criteria for those without a biopsy; (3) non-caseating granulomas in the biopsy for those with a biopsy [19]; and (4) initial presentation to the rheumatology clinic with a musculoskeletal symptom (arthritis, arthralgia, myalgia, and/or myositis).

To assess the presence of arthritis, a physical examination was conducted on the patients presenting with symptoms such as joint pain and swelling. The diagnosis of arthritis was based on detecting clinical findings such as pain, swelling, redness, increased local temperature, and/or joint limitation. In cases where the diagnosis of arthritis was uncertain, the confirmation was obtained by identifying effusion and/or an increased Doppler signal on joint ultrasonography. Detailed systemic examinations were performed on all of the patients. For the 47 patients diagnosed by our clinic, the diagnosis was based on systemic examination findings, elevated ACE levels, hypercalcemia/hypercalciuria, and the exclusion of other diseases. Some patients also underwent biopsies. In total, 63.27% (93 out of 147) had biopsies. The patients were thoroughly evaluated for differential diagnoses, including granulomatosis with polyangiitis; eosinophilic granulomatosis with polyangiitis; polyarteritis nodosa; tuberculosis; silicosis; foreign body reactions; and bacterial, fungal, or parasitic infections. Chest radiography was performed on all of the patients. Thoracic CT was indicated for those with findings suggestive of sarcoidosis, including bilateral or unilateral hilar enlargement, mediastinal enlargement, increased reticular densities, suspicious fibrotic recesses, honeycombing, or suspicious nodular opacities. For the patients with arthritis, the presence of other rheumatologic diseases was assessed to determine if the arthritis was attributable to sarcoidosis. Comprehensive neurological and ophthalmological examinations were also conducted, and the treatment plans were based on the affected organ/system.

### 2.2. Clinical and Laboratory Variables

The demographic, clinical, and initial laboratory characteristics of the patients were obtained from their medical records. These included age; gender; disease duration (months); organ and system involvement (lungs, lymph nodes, liver, spleen, eyes, breasts, joints, muscles, skin, neurological system, and lacrimal glands); comorbidities such as diabetes mellitus (DM), hypertension (HT), chronic obstructive pulmonary disease (COPD), asthma, chronic kidney disease (CKD), coronary artery disease (CAD), heart failure (HF), hypothyroidism, and osteoporosis; the presence of malignancy accompanying rheumatologic diseases; and the treatment agents used. The patients were classified as having monoarthritis (single joint), oligoarthritis (two to four joints), or polyarthritis (five or more joints). Both the active and past organ and system involvement was recorded separately based on the patient’s history. Scadding staging was performed using posteroanterior chest X-rays and thoracic CT scans, identifying hilar/mediastinal lymphadenopathy, interstitial lung involvement, and/or extrapulmonary involvement. Any accompanying rheumatologic diseases were diagnosed by correlating the clinical, laboratory, radiological, and, when necessary, pathological findings. The bronchoscopies and pulmonary function tests were performed by experienced pulmonologists and the echocardiography by experienced cardiologists. If bronchoalveolar lavage (BAL) was performed, the CD4/CD8 ratio was recorded; if echocardiography was performed, the ejection fraction (EF) and the pulmonary artery pressure (PAP) were noted; and if pulmonary function tests were conducted, the forced vital capacity (FVC), the forced expiratory volume in 1 s (FEV1), FEV1/FVC ratio, the diffusing capacity of the lungs for carbon monoxide adjusted (DLCO Adj), and the diffusing capacity of the lungs for carbon monoxide adjusted by divided alveolar volume (DLCO Adj/VA) were documented. The laboratory findings included the white blood count (WBC); the neutrophil, lymphocyte, hemoglobin, mean corpuscular volume (MCV), and platelet counts from the complete blood count; the urea, creatinine, calcium, phosphorus, alkaline phosphatase (ALP), parathyroid hormone (PTH), 25-hydroxyvitamin D, aspartate aminotransferase (AST), alanine transaminase (ALT), gamma-glutamyl transferase (GGT), C-reactive protein (CRP), and erythrocyte sedimentation rate (ESR) from the biochemical parameters; and the positivity of rheumatologic markers such as rheumatoid factors (RF), anti-citrullinated peptide antibodies (ACPA), and antinuclear antibody (ANA). The treatments used at the time of diagnosis and during the follow-up were also documented.

### 2.3. Statistical Analysis 

All of the statistical analyses were performed with SPSS (version 28.0; IBM Corporation, Armonk, NY, USA). The categorical variables are expressed as percentages (%) and numbers (*n*). Normally distributed quantitative data are presented as mean ± standard deviation (SD) [minimum, maximum]; non-normally distributed quantitative data are presented as median [interquartile range]. Whether there was a statistically significant difference between the groups of sarcoidosis patients with and without arthritis was determined using Pearson’s χ^2^ test for qualitative (categorical) data, the independent samples *t*-test for normally distributed quantitative data, and the Mann–Whitney U-test for non-normally distributed quantitative data. A univariable logistic regression analysis was initially conducted to identify the factors associated with the development of arthritis in the sarcoidosis patients. The significant variables in the univariable analysis were then included in a multivariable logistic regression model to adjust for potential confounders. This method facilitated the identification of the independent predictors of arthritis development, providing a more accurate assessment of the factors influencing this outcome. For all of the statistical analyses, *p* < 0.05 was accepted as the significance threshold.

This study was approved by a decision of the Clinical Research Ethics Committee of Bursa Uludag University dated 31 November 2023, with the number 2023-22/15. The study was performed in accordance with the Declaration of Helsinki.

## 3. Results

### 3.1. Patient Characteristics

Of the 181 patients, 34 were excluded because of incomplete data, resulting in 147 patients included in the study. Upon review, 47 patients were diagnosed with sarcoidosis by the rheumatology department, while 100 were diagnosed by other clinics (91 pulmonology, 8 dermatology, and 1 neurology) and referred to our rheumatology clinic due to musculoskeletal complaints. The mean age of the 147 patients was 56.02 ± 11.21 years, with an average disease duration of 134.33 ± 56.98 months. Of these patients, 86.40% were female. Pulmonary involvement was observed in 93.90% of the patients. All of the patients presented at least one musculoskeletal involvement. The other common involvements were the skin (21.20%), the eyes (14.30%), and peripheral lymphadenopathy (10.6%). 

Comorbid conditions were present in 59.90% of the patients, with hypertension (26.50%), diabetes mellitus (16.30%), and asthma (11.60%) being the most common. The frequencies of the rheumatic diseases accompanying sarcoidosis were as follows: RA, 6.12%; Sjögren’s syndrome, 2.70%; and idiopathic granulomatous mastitis, 1.40%. Additionally, ankylosing spondylitis (AS), mixed connective tissue disease, Behçet’s disease, and polymyalgia rheumatica (PMR) were each identified in one patient. Among the patients diagnosed with comorbid rheumatic diseases after sarcoidosis, four had RA, one had Sjögren’s, one had AS, and one had Behçet’s disease. Other patients had a pre-existing rheumatic diagnosis before their sarcoidosis diagnoseis. Malignancy was present in eleven patients (7.48%). Five of these (3.40%) were diagnosed with malignancy before sarcoidosis (one thyroid papillary carcinoma, one Kaposi’s sarcoma, one renal cell carcinoma, one breast cancer, and one prostate cancer). The remaining six patients (4.08%) were diagnosed with malignancy during their follow-up for sarcoidosis (two thyroid papillary carcinomas, one breast cancer, one endometrial cancer, one meningioma, and one cervical cancer).

The most frequently observed musculoskeletal symptoms associated with sarcoidosis were arthralgia (85.70%, 126 patients), myalgia (59.86%, 88 patients), and arthritis (30.61%, 45 patients). Among the 45 patients with sarcoidosis-associated arthritis, 10 (6.80%) had monoarthritis, 28 (19.05%) had oligoarthritis, and 7 (4.76%) had polyarthritis. The most commonly affected joints were the ankle (62.22%), the metacarpophalangeal joints (46.67%), the proximal interphalangeal joints (31.11%), and the knees (31.11%).

Of the 49 sarcoidosis patients with arthritis at presentation, ACPA positivity was found in 5. Three of these patients were diagnosed with RA. Two patients did not meet the RA criteria, and their arthritis was primarily attributed to sarcoidosis. One patient, initially without arthritis, developed ACPA positivity and met the RA criteria during the follow-up, leading to an RA diagnosis. Thus, 4 patients diagnosed with RA were excluded from the sarcoidosis-associated arthritis group, resulting in 45 patients diagnosed with sarcoidosis-associated arthritis. Six patients (4.02%) had Löfgren’s syndrome.

The most frequently used medication for sarcoidosis treatment was glucocorticoids, with 71.40% of patients using them. The median initial dose of glucocorticoids was 20 mg. Other commonly used treatments were methotrexate (29.30%), colchicine (21.80%), and hydroxychloroquine (21.10%). According to the Scadding staging, the most common stage was stage II (59.20%), followed by stage I (26.50%), stage 0 (6.80%), stage IV (4.10%), and stage III (3.40%) (Table 1 and Table 2).

### 3.2. Laboratory Measurements

In the laboratory examination of the patients, the frequency of microcytic anemia was 14.30% and that of normocytic anemia was 15.0%. The frequency of hypercalcemia was 13.60% and that of hypercalciuria was 10.20%. Elevated CRP levels were observed in 17.70% of the patients, and elevated ESR levels were observed in 26.50%. At diagnosis, 46.30% of the patients had elevated ACE levels, with a median ACE value of 50.30 U/L. ANA, ACPA, and RF positivity were 34.20%, 7.70%, and 4.50%, respectively. The ANA profile revealed Ro52 positivity in four patients, SS-A positivity in three, SS-B positivity in two, dsDNA positivity in one, Sm positivity in one, Scl-70 positivity in one, centromere B positivity in one, and PM-Scl positivity in one.

Among the 115 patients with available pulmonary function test results, the mean FVC (%) was 96.91 ± 22.16, the mean FEV1 (%) was 94.49 ± 20.16, the mean FEV1/FVC ratio was 84.58 ± 10.60, and the mean DLCO Adj (mL/mmHg/min) was 77.27 ± 23.30. In the 24 patients who underwent BAL with CD4 and CD8 analysis, the mean CD4/CD8 ratio was 3.70 ± 2.97. For the 34 patients who underwent transthoracic echocardiography, the mean ejection fraction (%) was 61.63 ± 3.74 and the mean pulmonary artery pressure (mmHg) was 29.44 ± 11.19 (Table 3).

### 3.3. Comparison of Patients with and Without Sarcoidosis-Associated Arthritis

A comparison was conducted between 45 sarcoidosis patients with arthritis and 102 sarcoidosis patients without arthritis The number of patients diagnosed with DM (*p* = 0.024), the number of patients with GGT levels above the laboratory reference range (*p* = 0.003), and the median GGT levels (*p* = 0.044) were statistically higher in the arthritis group compared to the non-arthritis group. Additionally, the DLCO Adj (mL/mmHg/min) value and the 25-hydroxyvitamin D levels (µg/L) were lower in the arthritis group than in the non-arthritis group (respectively, *p* = 0.039 and *p* = 0.002). No significant differences were observed in the other parameters (Table 4).

A univariable logistic regression analysis was initially conducted to identify the independent factors influencing the development of arthritis and to construct a predictive model for arthritis development. The presence of DM (*p* = 0.028, OR: 2.727, 95% CI: 1.115–6.668), ALT levels (U/L) (*p* = 0.027, OR: 1.034, 95% CI: 1.004–1.065), GGT levels (U/L) (*p* = 0.025, OR: 1.010, 95% CI: 1.001–1.019), and 25-hydroxyvitamin D levels (µg/L) (*p* = 0.025, OR: 0.937, 95% CI: 0.885–0.992) were significantly associated with the development of arthritis. In the multivariable analysis model, the presence of DM (*p* = 0.028, OR: 4.805, 95% CI: 1.025–22.518) and low 25-hydroxyvitamin D levels (*p* = 0.034, OR: 0.914, 95% CI: 0.841–0.993) classified arthritis development with 76.1% accuracy (Appendix A).

Biopsies were performed for diagnostic purposes in 93 of the 147 patients (63.27%). The most common biopsy sites were the mediastinal lymph nodes (52/93 patients, 55.91%), transbronchial biopsies (16/93, 17.20%), and skin biopsies (12/93, 12.90%). No significant difference was observed in the biopsy sites between the arthritis and non-arthritis groups (Table 5).

## 4. Discussion

Sarcoidosis is a multisystemic disease that can affect the musculoskeletal system, particularly the joints, muscles, bones, and blood vessels. In rare cases, musculoskeletal involvement can be the first clinical manifestation of sarcoidosis [20]. In our study, we examined the patients who presented to our clinic with any musculoskeletal complaints and were diagnosed with sarcoidosis, as well as the patients who were referred to our clinic after developing musculoskeletal complaints following a sarcoidosis diagnosis. By comparing the patients with and without arthritis, we highlighted the differences between the two groups, particularly noting that patients with diabetes mellitus and low vitamin D levels may be more prone to developing arthritis.

The mean age of our cohort was 56.02 ± 11.21 years, and 86.40% of the patients were female. The age of our patients was higher than that reported in some studies [21,22,23] and lower than that reported in others [23]. The high percentage of female patients was consistent with the literature [24,25,26]. The higher age of our patients may be explained by the later referral to the rheumatology clinic of the patients with pre-existing sarcoidosis who subsequently developed musculoskeletal complaints.

The frequency of pulmonary involvement in our patients was 93.90%, consistent with the literature [6,20,27]. The frequencies of extrapulmonary involvement, excluding arthralgia and myalgia, were as follows: arthritis, 30.61%; the skin, 21.20%; the eyes, 14.30%; and peripheral lymphadenopathy, 10.6%. The frequency of arthritis in our study was similar to the 15–25% reported in the literature [10,26]. Among the skin involvements, the frequencies of erythema nodosum (19.70%) and lupus pernio (1.40%) were lower in our study [28,29]. Uveitis was observed in 14.30% of the patients, with anterior uveitis (8.20%) and panuveitis (5.40%) being the most common types, consistent with the literature [29,30].

Comorbidities are significant factors in managing sarcoidosis. In our study, the most common comorbidities were HT and DM. The nature of sarcoidosis and the metabolic side effects of the glucocorticoids used in its treatment can contribute to the development and worsening of DM and HT, complicating disease management and reducing quality of life. Therefore, it is crucial to consider the existing comorbidities in patients undergoing treatment and to be vigilant about the potential development of DM and HT for effective disease management [31,32].

In our study, the most common rheumatic diseases accompanying sarcoidosis were RA (6.12%), Sjögren’s syndrome (2.70%), and granulomatous mastitis (1.36%). Rheumatic diseases such as RA and Sjögren’s syndrome are significant comorbidities that can accompany sarcoidosis, though their prevalence varies between studies. In a European cohort study by Brito-Zerón et al. (2021), the prevalence of RA in sarcoidosis patients was 0.52%, and that of Sjögren’s syndrome was 1.70%. These low prevalences align with a Taiwanese cohort study conducted by Wu et al. (2017), which reported 0.16% for RA and 1.54% for Sjögren’s syndrome [33,34]. In a Turkish study by Yıldız F. et al. (2016) involving 131 patients, the prevalence of RA was 2.29% and that of Sjögren’s syndrome was 0.76% [35]. The rheumatic diseases that can co-occur with sarcoidosis require careful consideration during diagnosis and treatment to account for these confounding factors.

The diagnosis of sarcoidosis is typically established through clinical and radiological findings, along with the histopathological demonstration of non-caseating granulomas. In our study, biopsies were performed for diagnostic purposes in 63.27% of the patients, with the most frequently used biopsy methods being mediastinal lymph node biopsy (55.91%), transbronchial biopsy (17.20%), and skin biopsy (12.90%). Rybicki et al. (1997) reported the use of mediastinal lymph node biopsy in 56%, transbronchial biopsy in 17%, and skin biopsy in 13% of sarcoidosis cases. Similarly, Baughman and Lower (2005) emphasized that mediastinal lymph node biopsy is the most commonly used diagnostic method [25,28].

In our study, the most common musculoskeletal findings in the patients diagnosed with sarcoidosis were arthralgia (85.70%), myalgia (59.86%), and arthritis (30.61%). These findings indicate that sarcoidosis can present with rheumatologic symptoms, leading patients to visit rheumatology clinics with these complaints. The literature reports a 15–25% arthritis frequency in patients with sarcoidosis [12,34,36]. In our study, the most frequently affected joints in the patients with arthritis were the ankle (62.22%), the metacarpophalangeal joints (46.67%), and the knees (31.11%). Similar findings have been reported in numerous studies, indicating that sarcoidosis-associated arthritis frequently involves the peripheral joints and requires careful evaluation in clinical management [12,22,36].

In our study, we compared 45 sarcoidosis patients with arthritis to 102 sarcoidosis patients without arthritis. The frequency of patients diagnosed with DM in the arthritis group (26.67%) was significantly higher than in the non-arthritis group (*p* = 0.024). Additionally, the frequency of patients with elevated GGT levels (*p* = 0.003) and the GGT levels themselves (*p* = 0.044) were significantly higher in the arthritis group. The literature also reports a relationship between DM, elevated liver enzymes, and the development of arthritis in sarcoidosis patients [33]. Several studies have explored the relationship between rheumatoid arthritis (RA) and diabetes mellitus (DM), as well as between osteoarthritis (OA) and DM. These studies suggest that chronic systemic inflammation and the metabolic effects of glucocorticoid therapy may contribute to the increased risk of DM observed in patients with these conditions [36,37]. These findings suggest that DM and liver dysfunction may trigger the inflammatory processes that increase the risk of developing arthritis.

The DLCO Adj (mL/mmHg/min) value was significantly lower in the arthritis group than in the non-arthritis group (*p* = 0.039). The lower DLCO Adj values may indicate that lung function is more severely affected in sarcoidosis patients with arthritis, highlighting the importance of comprehensive pulmonary evaluation in these patients.

In the multivariable logistic regression analysis conducted to determine the factors influencing the development of arthritis, the presence of DM (*p* = 0.028, OR: 4.805) and low levels of 25-hydroxyvitamin D (*p* = 0.034, OR: 0.914) were significantly associated with the development of arthritis. These findings suggest that DM may increase the risk of developing arthritis by triggering inflammatory processes, and low vitamin D levels may increase the arthritis risk owing to their effects on immune functions [32,37]. The literature also supports the association between DM, low vitamin D levels, and the development of arthritis [12,35,38].

Glucocorticoids are the most commonly preferred medications for treating sarcoidosis. In our study, 71.40% of the patients used glucocorticoids, with a median initial dose of 20 mg. Glucocorticoids effectively alleviate symptoms by suppressing inflammation and are the main treatment option for managing acute sarcoidosis attacks [39]. Additionally, 29.30% of the patients used methotrexate, 21.80% used colchicine, and 21.10% used hydroxychloroquine. Methotrexate is often used to reduce the side effects of long-term glucocorticoid use and as a steroid-sparing agent, while colchicine and hydroxychloroquine are more commonly used for skin and joint involvement. These medications play a significant role in the chronic management of sarcoidosis, particularly in treating arthritis [40,41].

Immunosuppressive drugs such as azathioprine (15.0%), mycophenolate mofetil (6.10%), and leflunomide (4.80%) are used in more resistant cases. These drugs are particularly important treatment options for patients with severe organ involvement and those who do not respond to conventional therapies [24,42]. Biological agents, especially infliximab, have been increasingly used to treat sarcoidosis in recent years. Infliximab, with its anti-TNF-α effect, suppresses inflammation and is effective in patients who do not respond to conventional therapies. Studies have shown that infliximab treatment improves symptoms and reduces disease activity in both pulmonary and extrapulmonary sarcoidosis [43,44]. In our study, two patients (1.40%) used infliximab. These patients responded to the treatment, providing evidence that biological agents are an effective treatment option, particularly in resistant cases. Infliximab is especially important in managing extrapulmonary involvement, cases that are resistant to conventional therapies, and severe systemic symptoms [43].

This study has several important strengths and limitations. The detailed examination of the clinical, demographic, and laboratory data from the 147 sarcoidosis patients who visited the rheumatology clinic between 2000 and 2024 and the comparison of the patients with and without arthritis enhance its value. Additionally, the roles of DM and low 25-hydroxyvitamin D levels in arthritis development were identified, and the accuracy of the findings was increased by using multivariable logistic regression analyses. This study shows that sarcoidosis can present with rheumatologic symptoms, leading to frequent visits to rheumatology clinics. Furthermore, the frequencies of rheumatologic diseases accompanying sarcoidosis were determined, highlighting the need to consider these diseases in diagnosis and treatment. 

However, our study has some limitations. Its retrospective design may have led to data deficiencies, and the single-center nature may limit the generalizability of the results. Additionally, factors such as patient adherence to treatment and treatment responses were not evaluated in detail. The analysis was constrained by incomplete clinical and laboratory data, and the distribution of joint involvement and other symptoms across the different disease stages was not fully examined. In this study, due to its retrospective design, ENA (extractable nuclear antigen antibodies) and ANCA (anti-neutrophil cytoplasmic antibodies) levels were not routinely measured. These biomarkers, however, play a crucial role in identifying coexisting autoimmune conditions in patients with sarcoidosis. Their inclusion in future prospective studies could provide a more comprehensive understanding of the potential relationships between sarcoidosis and overlapping rheumatologic diseases. Such investigations may offer valuable insights into the diagnostic and prognostic implications of these markers in this patient population. 

In our study, myalgia was identified as a common symptom in sarcoidosis patients (59.86%). However, myolytic enzymes, such as creatine kinase (CK) and lactate dehydrogenase (LDH), were not evaluated in this study. Measuring these enzymes could provide valuable insights into the potential relationship between myalgia and the underlying inflammatory or muscle damage mechanisms. We recommend that future research include the routine assessment of myolytic enzymes to better understand the pathophysiology of sarcoidosis-related myalgia. Such evaluations could contribute to improving the diagnostic and therapeutic approaches for this patient population. Another limitation is as follows. The relatively small sample size and further reduction in subgroup sizes may impact the statistical power and the generalizability of the findings. Additionally, the differences in the clinical characteristics among the patients could influence the results, and while multivariable logistic regression analyses were performed, residual confounding cannot be entirely excluded. These limitations underscore the need for future prospective studies with larger and more diverse cohorts to validate and extend our findings. Despite these limitations, this study is an important step in understanding the relationship between sarcoidosis and arthritis, highlighting the need for larger-scale prospective studies.

## 5. Conclusions

This study reveals significant clinical, demographic, and laboratory differences between sarcoidosis patients with and without arthritis. The sarcoidosis patients with arthritis exhibited a higher frequency of diabetes mellitus, lower levels of 25-hydroxyvitamin D, and elevated GGT levels, indicating a distinct clinical profile for this group. Additionally, the lower DLCO Adj values in the patients with arthritis suggest more severely affected lung functions, highlighting the importance of comprehensive pulmonary evaluation for this subgroup. These findings underscore the necessity of a multidisciplinary approach involving rheumatology, pulmonology, and other specialties in managing sarcoidosis. Larger prospective studies should be conducted to confirm these findings and better understand the development of arthritis in sarcoidosis.

## Figures and Tables

**Table 1 jcm-13-07563-t001:** Demographic and clinical characteristics of the patients (*n* = 147).

Patient Characteristics	Data
Age, mean ± SD (min., max.)	56.02 ± 11.21 (20.0, 83.0)
Gender, female/male (%)	86.40/13.60
Sarcoidosis disease duration (months), mean ± SD (min., max.)	134.33 ± 56.98 (2.0, 282.0)
Age at diagnosis (years), mean ± SD (min., max.)	44.86 ± 10.58 (14.0, 68.0)
Organ and system involvement	
Pulmonary involvement, % (*n*)	93.90 (138)
LAP, % (*n*)	91.8 (135)
Unilateral left, % (*n*)	0.70 (1)
Unilateral right, % (*n*)	4.80 (7)
Bilateral, % (*n*)	86.40 (127)
ILD, % (*n*)	66.67 (98)
Extrapulmonary involvement	
Arthralgia, % (*n*)	85.70 (126)
Myalgia, % (*n*)	59.86 (88)
Arthritis, % (*n*)	30.61 (45)
Monoarthritis, % (*n*)	6.80 (10)
Oligoarthritis, % (*n*)	19.05 (28)
Polyarthritis, % (*n*)	4.76 (7)
Skin, % (*n*)	21.20 (31)
Erythema nodosum, % (*n*)	19.70 (27)
Lupus pernio, % (*n*)	1.40 (2)
Others, % (*n*)	1.40 (2)
Eye, % (*n*)	14.3 (21)
Anterior uveitis, % (*n*)	8.20 (12)
Intermediate uveitis, % (*n*)	0.7 (1)
Panuveitis, % (*n*)	5.40 (8)
Peripheral lymph node, % (*n*)	10.90 (16)
Splenomegaly, % (*n*)	4.80 (7)
Liver, % (*n*)	3.40 (5)
Neurological, % (*n*)	2.0 (3)
Breast, % (*n*)	1.40 (2)
Lacrimal gland, % (*n*)	0.70 (1)
Cardiac, % (*n*)	0.70 (1)
Comorbidity, % (*n*)	59.90 (88)
HT	26.50 (39)
DM	16.30 (24)
Asthma	11.60 (17)
Osteoporosis	10.20 (15)
HL	7.50 (11)
Hypothyroidism	5.40 (8)
CAD	3.40 (5)
CKD	2.0 (3)
HF	1.40 (2)
COPD	0.70 (1)
Biopsy performed for diagnostic purposes, % (*n*) *	63.27 (93)
Mediastinal LAP, % (*n*)	55.91 (5)
Transbronchial biopsy, % (*n*)	17.20 (16)
Skin biopsy, % (*n*)	12.90 (12)
Lung parenchyma excision, % (*n*)	5.38 (5)
Axillary LAP, % (*n*)	4.30 (4)
Cervical LAP, % (*n*)	2.15 (2)
Liver, % (*n*)	1.08 (1)
Lacrimal gland, % (*n*)	1.08 (1)
Rheumatologic diseases accompanying sarcoidosis, % (*n*)	
RA	6.12 (9)
Sjögren’s	2.70 (4)
Granulomatosis mastitis	1.40 (2)
Others	2.72 (4)
Malignancy, % (*n*)	7.48 (11)
Distribution of drugs used in treatment, % (*n*)	
Corticosteroids	71.40 (105)
Methotrexate	29.30 (43)
Colchicine	21.80 (32)
Hydroxychloroquine	21.10 (31)
Azathioprine	15.0 (22)
Mycophenolate mofetil	6.10 (9)
Leflunomide	4.80 (7)
Sulfasalazine	4.10 (6)
Cyclosporine	2.70 (4)
Rituximab	2.70 (4)
Cyclophosphamide	2.70 (4)
Infliximab	1.40 (2)
Initial corticosteroid dose (mg), median (IQR)	20.0 (50.0)
Scadding stages, % (*n*)	
Stage 0	6.80 (10)
Stage I	26.50 (39)
Stage II	59.20 (87)
Stage III	3.40 (5)
Stage IV	4.10 (6)

SD: standard deviation; min.: minimum; max.: maximum; IQR: interquartile range; ILD: interstitial lung disease; LAP: lymphadenopathy; HT: hypertension; DM: diabetes mellitus; HL: hyperlipidemia; COPD: chronic obstructive pulmonary disease; CAD: coronary artery disease; CKD: chronic kidney disease; HF: heart failure; RA: rheumatoid arthritis; *: the percentages were calculated based on 93 patients.

**Table 2 jcm-13-07563-t002:** Joint involvement regions in the group of patients with arthritis (*n* = 45).

Regions	% (*n*)
Ankle	62.22 (28)
MCP	46.67 (21)
PIP	31.11 (14)
Knee	31.11 (14)
Wrist	28.89 (13)
Elbow	13.33 (6)
MTP	11.11 (5)
DIP	6.67 (3)
Shoulder	4.44 (2)

MCP: metacarpophalangeal joint; PIP: proximal interphalangeal joint; MTP: metatarsophalangeal joint; DIP: distal interphalangeal joint.

**Table 3 jcm-13-07563-t003:** Laboratory characteristics of the patients.

Patient Characteristics	Data
Complete blood count (*n* = 147)	
WBC (10^9^/mL), median (IQR)	7.220 (3.190)
Neutrophil (10^9^/mL), median (IQR)	4.480 (2.910)
Lymphocyte (10^9^/mL), mean ± SD (min., max.)	1.98 ± 0.79 (5.28, 4.73)
Hemoglobin (g/dL), mean ± SD (min., max.)	12.66 ± 1.39 (9.26, 16.20)
MCV (fL), mean ± SD (min., max.)	82.46 ± 6.18 (60.40, 94.30)
Platelet (10^3^/mL), median (IQR)	280.0 (92.0)
Biochemical tests (*n* = 147)	
Calcium (8.40–10.20 mg/dL), mean ± SD (min., max.)	9.33 ± 0.62 (6.50, 11.40)
Phosphorous (2.30–4.70 mg/dL), mean ± SD (min., max.)	3.56 ± 0.92 (1.40, 9.0)
Parathormone (15.0–68.30 ng/L), mean ± SD (min., max.)	128.40 ± 328.62 (7.8, 2647.0)
25-hydroxyvitamin D (20–50 µg/L), median (IQR)	12.32 (11.48)
Hypercalcemia, % (*n*)	13.60 (20)
Hypercalciuria, % (*n*)	10.20 (15)
GGT, (U/L), median (IQR)	26.0 (36.50)
Elevated GGT, % (*n*)	21.09 (31)
ALP (U/L), median (IQR)	80.50 (39.50)
Elevated ALP, % (*n*)	6.10 (9)
AST (U/L), median (IQR)	18.0 (8.0)
Elevated AST, % (*n*)	8.20 (12)
ALT (U/L), median (IQR)	17.0 (11.0)
Elevated ALT, % (*n*)	2.70 (4.0)
CRP (mg/dL), median (IQR)	1.08 (3.36)
Elevated CRP, % (*n*)	17.70 (26)
ESR (mm/hour), median (IQR)	26.50 (32.25)
Elevated ESR, % (*n*)	60.50 (89)
ACE (U/L), median (IQR)	50.30 (55.0)
Elevated serum ACE, % (*n*)	46.30 (68)
Autoantibodies, % (*n*)	28.57 (42)
RF positivity (*n* = 112)	4.50 (5)
ACPA positivity (*n* = 78)	7.70 (6)
ANA positivity (*n* = 117)	34.20 (40)
PFTs (*n* = 115), mean ± SD (min., max.)	
FVC (L)	2.76 ± 0.88 (0.08, 5.36)
FVC (%)	96.91 ± 22.16 (26.0, 147.0)
FEV1 (L)	2.29 ± 0.69 (0.82, 4.56)
FEV1 (%)	94.49 ± 20.16 (46.0, 145.0)
FEV1/FVC (%)	84.58 ± 10.60 (59.0, 126.0)
DLCO Adj (mL/mmHg/min)	77.27 ± 23.30 (20.90, 169.0)
DLCO/VA (mL/mmHg/min/L)	86.07 ± 22.39 (1.94, 137.0)
Echocardiography (*n* = 34), mean ± SD (Min., Max.)	
EF (%)	61.63 ± 3.74 (53.0, 70.0)
PAP (mmHg)	29.44 ± 11.19 (18.0, 75.0)
BAL (*n* = 24), mean ± SD (min., max.)	
CD4/CD8 ratio	3.70 ± 2.97 (0.36, 10.40)

SD: standard deviation; min.: minimum; max.: maximum; IQR: interquartile range; L: Liter; mL: milliliter; dL: deciliter; g: gram; µg: microgram; ng: nanogram; mmHg: millimeters of mercury; fL: femtoliter; WBC: white blood count (10^9^/L); MCV: mean corpuscular volume (N: 81.1–96 f L); AST: aspartate aminotransferase (N: 5–34 U/L); ALT: alanine aminotransferase (N: 8–55 U/L); ALP: alkaline phosphatase (40–150 U/L); GGT: gamma-glutamyl transferase (12–64 U/L); CRP: C-reactive protein (0–5 mg/L); ESR: erythrocyte sedimentation rate 1st hour (2–20 mm/hour); ACE: angiotensin-converting enzyme (8–52 U/L); RF: rheumatoid factor (5.5–30 U/mL); ACPA: anti-citrullinated protein antibody; ANA: antinuclear antibody; FVC: forced vital capacity; FEV1: forced expiratory volume in 1 s; DLCO: diffusing capacity of the lungs for carbon monoxide; EF: ejection fraction; PAP: pulmonary artery pressure; BAL: bronchoalveolar lavage; CD: cluster of differentiation.

**Table 4 jcm-13-07563-t004:** Comparison of characteristics between sarcoidosis patients with and without arthritis.

	Patients with Arthritis (*n* = 45)	Patients Without Arthritis (*n* = 102)	*p*-Value *
Gender, F/M (*n*)	36/9	91/11	0.133
Age, mean ± SD	58.16 ± 11.06	54.96 ± 11.18	0.102
Sarcoidosis disease duration (months), mean ± SD	143.37 ± 51.60	129.81 ± 59.22	0.175
Age at diagnosis (years), mean ± SD	46.27 ± 10.21	44.15 ± 10.74	0.255
Comorbidity, % (*n*)	66.67 (30)	56.86 (58)	0.268
DM, % (*n*)	12 (26.67)	12 (11.76)	** *0.024* **
Hypercalciuria, % (*n*)	13.33 (6)	8.82 (9)	0.405
Hypercalcemia, % (*n*)	8.89 (4)	15.69 (16)	0.268
Pulmonary involvement, % (*n*)	93.33 (42)	94.12 (96)	0.855
ILD, % (*n*)	62.22 (28)	64.71 (66)	0.773
Hilar/mediastinal LAP	93.33 (42)	91.18 (93)	0.645
Liver involvement, % (*n*)	4.44 (2)	2.94 (3)	0.643
Eye involvement, % (*n*)	8.89 (4)	16.67 (17)	0.214
Peripheral LAP, % (*n*)	17.78 (8)	7.84 (8)	0.075
Skin involvement, % (*n*)	22.22 (10)	20.59 (21)	0.823
Neurological involvement, % (*n*)	2.22 (1)	1.96 (2)	0.418
Splenomegaly, % (*n*)	6.67 (3)	3.92 (4)	0.471
Erythema nodosum, % (*n*)	26.67 (12)	17 (16.67)	0.160
Elevated CRP, % (*n*)	24.44 (11)	14.71 (15)	0.154
CRP (mg/dL), median (IQR)	1.22 (4.84)	1.07 (2.64)	0.246
Elevated ESR, % (*n*)	71.11 (32)	55.88 (57)	0.082
ESR (mm/hour), median (IQR)	31.0 (25.50)	25.0 (36.0)	0.272
Elevated serum ACE, % (*n*)	44.44 (20)	47.06 (48)	0.770
Serum ACE, (U/L), median (IQR)	49.0 (54.45)	52.0 (54.35)	0.837
RF positivity, % (*n*)	2.56 (1) (*n* = 39)	5.48 (4) (*n* = 73)	0.579
ACPA positivity, % (*n*)	6.90 (2) (*n* = 29)	8.16 (4) (*n* = 49)	0.944
ANA positivity, % (*n*)	35.90 (14) (*n* = 39)	33.33 (26) (*n* = 78)	0.571
AST, (U/L), median (IQR)	19.0 (12.50)	18.0 (7.25)	0.469
ALT, (U/L), median (IQR)	19.0 (37.0)	16.50 (9.0)	0.118
GGT, (U/L), median (IQR)	38.0 (58.0)	24.0 (21.0)	** *0.044* **
25-hydroxyvitamin D (µg/L), median (IQR)	12.15 (4.30)	14.10 (15.20)	** *0.002* **
Elevated GGT, % (*n*)	37.78 (17)	13.73 (14)	** *0.003* **
Elevated ALP, % (*n*)	6.67 (3)	5.88 (6)	0.906
Elevated ALT, % (*n*)	6.67 (3)	0.98 (1)	0.051
Elevated AST, % (*n*)	13.33 (6)	5.89 (6)	0.128
DLCO Adj (mL/mmHg/min.), mean ± SD	70.53 ± 25.87	80.59 ± 21.35	** *0.039* **
Scadding stages, % (*n*)			0.869
Stage 0	8.20 (4)	6.10 (6)	
Stage I	22.40 (11)	28.60 (28)	
Stage II	63.30 (31)	57.10 (56)	
Stage III	2.0 (1)	4.10 (4)	
Stage IV	4.10 (2)	4.10 (4)	

SD: standard deviation; IQR: interquartile range; F: female; M: male; ILD: interstitial lung disease; LAP: lymphadenopathy; CRP: C-reactive protein; ESR: erythrocyte sedimentation rate 1st hour; ACE: angiotensin-converting enzyme; RF: rheumatoid factor; ACPA: anti-citrullinated protein antibody; ANA: antinuclear antibody; GGT: gamma-glutamyl transferase; ALT: alanine aminotransferase; DLCO: diffusing capacity of the lungs for carbon monoxide; *: statistical significance was defined as *p* < 0.05.

**Table 5 jcm-13-07563-t005:** Sites of biopsies performed for diagnostic purposes (*n* = 93).

	Patients with Arthritis (*n* = 28)	Patients Without Arthritis (*n* = 65)	*p*-Value *
Mediastinal LAP, % (*n*)	50.0 (14)	58.46 (38)	0.473
Transbronchial biopsy, % (*n*)	17.86 (5)	16.92 (11)	0.953
Cervical LAP, % (*n*)	3.57 (1)	1.54 (1)	0.549
Axillary LAP, % (*n*)	3.57 (1)	4.62 (3)	0.805
Skin biopsy, % (*n*)	21.43 (6)	9.23 (6)	0.128
Liver, % (*n*)	0	1.54 (1)	0.505
Lung parenchyma excision, % (*n*)	3.57 (1)	6.15 (4)	0.600
Lacrimal gland, % (*n*)	0	1 (1.54)	0.505

LAP: lymphadenopathy; *: statistical significance was defined as *p* < 0.05.

## Data Availability

The datasets generated and/or analyzed during the current study are available from the corresponding author upon reasonable request.

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
