# Peer review of "Rheumatologic Perspectives on Sarcoidosis: Predicting Sarcoidosis-Associated Arthritis Through Comprehensive Clinical and Laboratory Assessment"

_jcm, 2024, doi:10.3390/jcm13247563_

Round 1

Reviewer 1 Report

Comments and Suggestions for Authors

This manuscript is a retrospective and single center study to evaluate the characteristics of sarcoidosis patients with musculoskeletal symptoms. The authors showed patients with sarcoidosis-associated arthritis exhibit a higher frequency of diabetes mellitus, lower levels of 25-hydroxyvitamin D, and elevated GGT levels. Additionally, lower DLCO values in patients with arthritis indicate a more severe impact on pulmonary function, underscoring the importance of comprehensive pulmonary evaluation in this subgroup.

While there are some interesting points in the results, the description of discussion is insufficient.

Comments;

1.Regarding the investigation of complications, did you use questionnaires or other methods? Or was the investigation based on the medical record entries of the physician in charge? Depending on the method of investigation, the results showing an association with arthritis may be unreliable.

2.What were the characteristics of the patients with and without arthralgia and myalgia other than arthritis, and is there a reason why you focused only on arthritis?

3.In the discussion, you mention diabetes being associated with arthritis, but have other diseases presenting with arthritis also been shown to be associated with diabetes?

4. Overall, the discussion is redundant and seems unnecessary in some areas. I request a description of the characteristics of sarcoidosis patients with musculoskeletal symptoms derived from the present results, focusing on the characteristics associated with arthritis.

Author Response

Comments 1: Regarding the investigation of complications, did you use questionnaires or other methods? Or was the investigation based on the medical record entries of the physician in charge? Depending on the method of investigation, the results showing an association with arthritis may be unreliable.

Answer 1: Thank you for your valuable comment. The investigation of complications in our study was based primarily on the medical record entries of the physicians in charge. These records included detailed clinical notes, laboratory results, and imaging findings. While this approach ensures consistency with the retrospective design, we acknowledge the potential limitations associated with relying solely on medical records, such as variability in documentation practices and possible underreporting of some symptoms. We did not use patient questionnaires or other methods for data collection, which may limit the comprehensiveness of symptom documentation.

However, in our clinic, especially in the diagnosis of arthritis, physical examination findings defining arthritis together with patient complaints and, if necessary, ultrasonographic examination of the relevant joint are used. No patient with missing data or suspicion about the clinical findings related to the diagnosis was included in the study. Other system and organ findings were obtained by examining the patient records in the consultation notes of the relevant clinicians. This issue is described in detail in the method section and the limitations of the study are also described.

Comments 2: What were the characteristics of the patients with and without arthralgia and myalgia other than arthritis, and is there a reason why you focused only on arthritis?

Answer 2: In our study, the characteristics of patients with and without arthralgia and myalgia were recorded as part of the musculoskeletal symptoms; however, the focus was primarily on arthritis due to its distinct clinical significance and potential to influence treatment strategies.

Patients with arthralgia and myalgia presented with a range of symptoms, but these were generally less specific compared to arthritis. Arthralgia (present in 85.7% of patients) and myalgia (present in 59.9%) were often reported without clear objective findings, making them more challenging to analyze and interpret in terms of their clinical impact. In contrast, arthritis, with its defined diagnostic criteria (such as joint swelling, pain, and ultrasonographic findings), allowed for more objective evaluation and statistical analysis.

The decision to focus on arthritis was also influenced by its more direct implications for systemic disease burden, as it is associated with distinct laboratory and clinical findings, such as higher frequencies of diabetes mellitus, lower vitamin D levels, and elevated GGT levels in our cohort. These differences underscore the importance of arthritis in the overall management of sarcoidosis patients, and thus, it became the primary focus of this study. We appreciate your inquiry and have clarified this rationale in the discussion section of the revised manuscript.

Comments 3: In the discussion, you mention diabetes being associated with arthritis, but have other diseases presenting with arthritis also been shown to be associated with diabetes?

Answer 3: Thank you for your valuable comment regarding the association of diabetes mellitus (DM) with other diseases presenting with arthritis. To address this, the following statement has been added to the discussion section of the revised manuscript:

Line 372- 376:

"Several studies have explored the relationship between rheumatoid arthritis (RA) and diabetes mellitus (DM), as well as between osteoarthritis (OA) and DM. These studies suggest that chronic systemic inflammation and the metabolic effects of glucocorticoid therapy may contribute to the increased risk of DM observed in patients with these conditions [32, 33]."

Additionally, the reference list has been updated to include the following studies:

  1. Jiang, P., Li, H., & Li, X. (2015) Diabetes mellitus risk factors in rheumatoid arthritis: a systematic review and meta-analysis. Clinical and experimental rheumatology. 33(1): 115–121.
  2. Louati, K., Vidal, C., Berenbaum, F. et al (2015) Association between diabetes mellitus and osteoarthritis: systematic literature review and meta-analysis. RMD open 1(1): e000077. https://doi.org/10.1136/rmdopen-2015-000077.

Comments 4: Thank you very much for your valuable comments.

In the final paragraph of the introduction section of our study, we outlined our objectives as follows:
Firstly, we aimed to evaluate the demographic, clinical, and laboratory characteristics of patients diagnosed with sarcoidosis who developed musculoskeletal symptoms and patients with musculoskeletal symptoms who were diagnosed with sarcoidosis as a result of diagnostic investigations. This study particularly sought to investigate the relationship between the development of arthritis and various laboratory parameters, including vitamin D levels, liver enzyme levels, and ACE levels, as well as clinical features and comorbid conditions. A comparative analysis was conducted between patients with sarcoidosis-associated arthritis and those without, focusing on clinical and laboratory similarities and distinctions.

Additionally, we aimed to emphasize that not all arthritis cases observed in sarcoidosis patients are attributable to the underlying disease and to explore the factors contributing to the development of sarcoidosis-related arthritis from a rheumatologic perspective. Furthermore, we presented our experience regarding the role of biopsy in diagnosing sarcoidosis within rheumatological practice.

Therefore, our study aimed both to provide a comprehensive comparison of the characteristics and data between the two groups and to highlight the factors that may predict the development of arthritis in sarcoidosis patients.

Thank you again for your constructive feedback, which has helped us clarify the objectives and significance of our work.

Reviewer 2 Report

Comments and Suggestions for Authors

The topic is interesting and the paper is quite well written. I have some comments:

1) Abstract. Methods: This retrospective study analyzed 147 sarcoidosis patients from 2000 to 2024, categorized by the presence (n=45) or absence (n=102) of arthritis. Demographic, clinical, and laboratory data, including biopsy results, were collected and compared. Results: The mean age was 56.02±11.21 years, with a mean disease duration of 134.33 ± 56.98 months. Females constituted 86.4% of the cohort. All patients presented musculoskeletal involvement. Pulmonary involvement was present in 93.7% of cases, and extrapulmonary involvement included skin (21.20%), eyes (14.30%), and peripheral lymphadenopathy (10.6%). Methotrexate was the most common treatment after corticosteroids. In the arthritis group, diabetes mellitus was more frequent (p=0.024), GGT levels were higher (p=0.044), and 25-hydroxyvitamin D levels (p=0.002) and DLCOAdj (p=0.039) were lower. Multivariable regression showed diabetes mellitus (p=0.028, OR: 4.805, 95% CI: 1.025–22.518) and low 25-hydroxyvitamin D levels (p=0.034, OR: 0.914, 95% CI: 0.841–0.993) as factors influencing arthritis development. Other parameters showed no significant differences. Conclusions: This study identified significant clinical, demographic, and laboratory differences between sarcoidosis patients with and without arthritis. Patients with sarcoidosis-associated arthritis exhibit a higher frequency of diabetes mellitus, lower levels of 25-hydroxyvitamin D, and elevated GGT levels. Additionally, lower DLCO values in patients with arthritis indicate a more severe impact on pulmonary function, underscoring the importance of comprehensive pulmonary evaluation in this subgroup. I suggest to undreline in the results the statistically significant values to support the conclusions.

2) 1. Introduction 40 Sarcoidosis is a rare systemic disease of unknown etiology, characterized histopatho-41 logically by non-caseating granulomas [1,2]. It can involve multiple organs and systems, ... I suuedt to improve this paragraph and add some most important references, such as:

1- Is YouTube a sufficient source of information on Sarcoidosis?. Respir Res. 2024;25(1):334. Published 2024 Sep 9. doi:10.1186/s12931-024-02956-2

2- Pulmonary Sarcoidosis and Immune Dysregulation: A Pilot Study on Possible Correlation. Diagnostics (Basel). 2023;13(18):2899. Published 2023 Sep 11. doi:10.3390/diagnostics13182899

3) Sarcoidosis can mimic and coexist with diseases such as 85 rheumatoid arthritis (RA), ankylosing spondylitis (AS), Sjögren's syndrome, and vascu-86 litis. Therefore, rheumatologists must consider sarcoidosis in their differential diagnosis. This is an important topic I suggest to support the information with some recent references.

4)  Laboratory findings included white blood count (WBC), neutrophil, lym-154 phocyte, hemoglobin, mean corpuscular volume (MCV), and platelet counts from the 155 complete blood count; urea, creatinine, calcium, phosphorus, alkaline phosphatase (ALP), 156 parathyroid hormone (PTH), 25-hydroxyvitamin D, aspartate aminotransferase (AST), al-157 anine transaminase (ALT), gamma-glutamyl transferase (GGT), C-reactive protein (CRP), 158 and erythrocyte sedimentation rate (ESR) from biochemical parameters; and the positivity 159 of rheumatologic markers such as rheumatoid factors (RF), anti-citrullinated peptide an-160 tibodies (ACPA), and antinuclear antibody (ANA). The treatments used at the time of 161 diagnosis and during follow-up were also documented. Can you specify if you dosed the ENAand ANCA?

5) The most frequently observed musculoskeletal symptoms associated with sarcoido-208 sis were arthralgia (85.70%, 126 patients), myalgia (59.86%, 88 patients), and arthritis 209 (30.61%, 45 patients). Among the 45 patients with sarcoidosis-associated arthritis, 10 210 (6.80%) had monoarthritis, 28 (19.05%) had oligoarthritis, and 7 (4.76%) had polyarthritis. 211 The most commonly affected joints were the ankle (62.22%), metacarpophalangeal joints 212 (46.67%), proximal interphalangeal joints (31.11%), and knees (31.11%). Was myalgia associated with increased myolytic enzymes?

6) This study has several important strengths and limitations. The detailed examination 409 of clinical, demographic, and laboratory data from 147 sarcoidosis patients who visited 410 the rheumatology clinic between 2000 and 2024 and the comparison of patients with and 411 without arthritis enhance its value. Additionally, the roles of DM and low 25-hydroxyvit-412 amin D levels in arthritis development were identified, and the accuracy of the findings 413 was increased using multivariable logistic regression analyses. This study shows that sar-414 coidosis can present with rheumatologic symptoms, leading to frequent visits to rheuma-415 tology clinics. Furthermore, the frequencies of rheumatologic diseases accompanying sar-416 coidosis were determined, highlighting the need to consider these diseases in diagnosis 417 and treatment. I think it is fair to point out that the group was limited and the subgroups were further reduced and this can affect the statistical analysis. Furthermore, differences in clinical characteristics can influence the results.

Author Response

Comments 1: 1) Abstract. Methods: This retrospective study analyzed 147 sarcoidosis patients from 2000 to 2024, categorized by the presence (n=45) or absence (n=102) of arthritis. Demographic, clinical, and laboratory data, including biopsy results, were collected and compared. Results: The mean age was 56.02±11.21 years, with a mean disease duration of 134.33 ± 56.98 months. Females constituted 86.4% of the cohort. All patients presented musculoskeletal involvement. Pulmonary involvement was present in 93.7% of cases, and extrapulmonary involvement included skin (21.20%), eyes (14.30%), and peripheral lymphadenopathy (10.6%). Methotrexate was the most common treatment after corticosteroids. In the arthritis group, diabetes mellitus was more frequent (p=0.024), GGT levels were higher (p=0.044), and 25-hydroxyvitamin D levels (p=0.002) and DLCOAdj (p=0.039) were lower. Multivariable regression showed diabetes mellitus (p=0.028, OR: 4.805, 95% CI: 1.025–22.518) and low 25-hydroxyvitamin D levels (p=0.034, OR: 0.914, 95% CI: 0.841–0.993) as factors influencing arthritis development. Other parameters showed no significant differences. Conclusions: This study identified significant clinical, demographic, and laboratory differences between sarcoidosis patients with and without arthritis. Patients with sarcoidosis-associated arthritis exhibit a higher frequency of diabetes mellitus, lower levels of 25-hydroxyvitamin D, and elevated GGT levels. Additionally, lower DLCO values in patients with arthritis indicate a more severe impact on pulmonary function, underscoring the importance of comprehensive pulmonary evaluation in this subgroup. I suggest to undreline in the results the statistically significant values to support the conclusions.

Answer 1: Thank you for your valuable comment. In the revised abstract, we have highlighted the statistically significant p-values by making them bold to ensure clarity and emphasize their importance in supporting the study's conclusions.

Comments 2: 1. Introduction 40 Sarcoidosis is a rare systemic disease of unknown etiology, characterized histopatho-41 logically by non-caseating granulomas [1,2]. It can involve multiple organs and systems, ... I suuedt to improve this paragraph and add some most important references, such as:

1- Is YouTube a sufficient source of information on Sarcoidosis?. Respir Res. 2024;25(1):334. Published 2024 Sep 9. doi:10.1186/s12931-024-02956-2

2- Pulmonary Sarcoidosis and Immune Dysregulation: A Pilot Study on Possible Correlation. Diagnostics (Basel). 2023;13(18):2899. Published 2023 Sep 11. doi:10.3390/diagnostics13182899

Answer 2: Thank you for your insightful comment.  In the revised manuscript, we have added the following reference to the introduction section to provide additional context and strengthen the discussion on immune dysregulation in sarcoidosis:

*Cifaldi, R., Salton, F., Confalonieri, P., Trotta, L., Barbieri, M., Ruggero, L., Valeri, G., Pozzan, R., Della Porta, R., Kodric, M., Baratella, E., Bellan, M., Lerda, S., Hughes, M., Confalonieri, M., Cova, M. A., Gandin, I., Mondini, L., & Ruaro, B. (2023). Pulmonary Sarcoidosis and Immune Dysregulation: A Pilot Study on Possible Correlation. Diagnostics (Basel, Switzerland), 13(18), 2899. https://doi.org/10.3390/diagnostics13182899

Comments 3: Sarcoidosis can mimic and coexist with diseases such as 85 rheumatoid arthritis (RA), ankylosing spondylitis (AS), Sjögren's syndrome, and vascu-86 litis. Therefore, rheumatologists must consider sarcoidosis in their differential diagnosis. This is an important topic I suggest to support the information with some recent references.

Answer 3: Thank you for your valuable comment and contribution. With reference 10, the following reference has been added to number 18.

  1. Kobak S, Sever F, Sivrikoz ON, Orman M (2014) Sarcoidois: is it only a mimicker of primary rheumatic disease? A single center experience. Therapeutic advances in musculoskeletal disease 6(1):3–7. https://doi.org/10.1177/1759720X13511197

  1. Korsten P, Tampe B, Konig MF et al (2018) Sarcoidosis and autoimmune diseases: differences, similarities and overlaps. Curr Opin Pulm Med 24(5):504-512. doi:10.1097/MCP.0000000000000500

Comments 4:  Laboratory findings included white blood count (WBC), neutrophil, lym-154 phocyte, hemoglobin, mean corpuscular volume (MCV), and platelet counts from the 155 complete blood count; urea, creatinine, calcium, phosphorus, alkaline phosphatase (ALP), 156 parathyroid hormone (PTH), 25-hydroxyvitamin D, aspartate aminotransferase (AST), al-157 anine transaminase (ALT), gamma-glutamyl transferase (GGT), C-reactive protein (CRP), 158 and erythrocyte sedimentation rate (ESR) from biochemical parameters; and the positivity 159 of rheumatologic markers such as rheumatoid factors (RF), anti-citrullinated peptide an-160 tibodies (ACPA), and antinuclear antibody (ANA). The treatments used at the time of 161 diagnosis and during follow-up were also documented. Can you specify if you dosed the ENAand ANCA?

Answer 4: Thank you for your valuable comment regarding the measurement of ENA and ANCA in our study. We acknowledge the importance of these biomarkers in identifying coexisting autoimmune conditions in sarcoidosis patients.

In the revised manuscript, we have added the following statement to the discussion section to address this point:

"In this study, due to its retrospective design, ENA (extractable nuclear antigen antibodies) and ANCA (anti-neutrophil cytoplasmic antibodies) levels were not routinely measured. These biomarkers, however, play a crucial role in identifying coexisting autoimmune conditions in patients with sarcoidosis. Their inclusion in future prospective studies could provide a more comprehensive understanding of the potential relationships between sarcoidosis and overlapping rheumatologic diseases. Such investigations may offer valuable insights into the diagnostic and prognostic implications of these markers in this patient population."

We believe this addition clarifies the limitation of our study while suggesting a direction for future research. Thank you for highlighting this important point, which has helped us improve the quality of our manuscript.

Comments 5: Thank you for your insightful question regarding the potential association between myalgia and myolytic enzyme levels in sarcoidosis patients.

In our study, we did not evaluate myolytic enzymes, such as creatine kinase (CK) and lactate dehydrogenase (LDH), due to the retrospective design and the absence of routine measurements for these biomarkers in our cohort. To address this, we have added the following statement to the discussion section:

*"In our study, myalgia was identified as a common symptom in sarcoidosis patients (59.86%). However, myolytic enzymes, such as creatine kinase (CK) and lactate dehydrogenase (LDH), were not evaluated in this study. Measuring these enzymes could provide valuable insights into the potential relationship between myalgia and underlying inflammatory or muscle damage mechanisms.

This limitation stems from the retrospective design of our study. We recommend that future research include routine assessment of myolytic enzymes to better understand the pathophysiology of sarcoidosis-related myalgia. Such evaluations could contribute to improving diagnostic and therapeutic approaches for this patient population."*

We believe this addition clarifies the limitation and provides direction for future research in this area. Thank you for highlighting this important aspect, which has helped us refine our discussion and improve the manuscript.

Comments 6:  This study has several important strengths and limitations. The detailed examination 409 of clinical, demographic, and laboratory data from 147 sarcoidosis patients who visited 410 the rheumatology clinic between 2000 and 2024 and the comparison of patients with and 411 without arthritis enhance its value. Additionally, the roles of DM and low 25-hydroxyvit-412 amin D levels in arthritis development were identified, and the accuracy of the findings 413 was increased using multivariable logistic regression analyses. This study shows that sar-414 coidosis can present with rheumatologic symptoms, leading to frequent visits to rheuma-415 tology clinics. Furthermore, the frequencies of rheumatologic diseases accompanying sar-416 coidosis were determined, highlighting the need to consider these diseases in diagnosis 417 and treatment. I think it is fair to point out that the group was limited and the subgroups were further reduced and this can affect the statistical analysis. Furthermore, differences in clinical characteristics can influence the results.

Answer 6: Thank you for your thoughtful feedback regarding the strengths and limitations of our study. We acknowledge the limitations you have highlighted and appreciate the opportunity to address these points.

  1. Sample Size and Subgroup Reduction:
    We agree that the relatively small sample size, especially after subgroup stratification, is a limitation of our study. This limitation has been explicitly acknowledged in the discussion section. The reduced number of patients in certain subgroups may affect the power of statistical analyses and the generalizability of the findings. Future studies with larger and more diverse cohorts are necessary to confirm and expand upon our results.
  2. Differences in Clinical Characteristics:
    We also recognize that variations in clinical characteristics among patients could influence the results. While multivariable logistic regression analyses were performed to control for confounding factors, residual confounding cannot be completely ruled out due to the retrospective nature of the study. This limitation has been addressed in the discussion, and we recommend future prospective studies to better account for these differences.

We have added the following statement to the discussion section to reflect these limitations:

" Another limitation is as follows. The relatively small sample size and further reduction in subgroup sizes may impact the statistical power and the generalizability of the findings. Additionally, differences in clinical characteristics among patients could influence the results, and while multivariable logistic regression analyses were performed, residual confounding cannot be entirely excluded. These limitations underscore the need for future prospective studies with larger and more diverse cohorts to validate and extend our findings."

Thank you again for your constructive critique, which has helped us enhance the clarity and transparency of our manuscript.

Round 2

Reviewer 1 Report

Comments and Suggestions for Authors

The authors have provided appropriate responses to the points of concern.

Reviewer 2 Report

Comments and Suggestions for Authors The authors adequately answered my comments. They edited the manuscript and took my suggestions into account. In my opinion this improved the manuscript. I have no further comments.